# CD39$^+$PD-1$^+$CD8$^+$ T cells mediate metastatic dormancy in breast cancer

Paulino Tallón de Lara [1,2,6,9], Héctor Castañón[1,2,9], Marijne Vermeer [1,2], Nicolás Núñez [1,2], Karina Silina [1,2], Bettina Sobottka[3], Joaquín Urdinez[4,7], Virginia Cecconi[1,2], Hideo Yagita[5], Farkhondeh Movahedian Attar[1,2], Stefanie Hiltbrunner [1,2,8], Isabelle Glarner[1,2], Holger Moch[2,3], Sònia Tugues[1,2], Burkhard Becher [1,2] & Maries van den Broek [1,2✉]

Some breast tumors metastasize aggressively whereas others remain dormant for years. The mechanism governing metastatic dormancy remains largely unknown. Through high-parametric single-cell mapping in mice, we identify a discrete population of CD39$^+$PD-1$^+$ CD8$^+$ T cells in primary tumors and in dormant metastasis, which is hardly found in aggressively metastasizing tumors. Using blocking antibodies, we find that dormancy depends on TNF$\alpha$ and IFN$\gamma$. Immunotherapy reduces the number of dormant cancer cells in the lungs. Adoptive transfer of purified CD39$^+$PD-1$^+$CD8$^+$ T cells prevents metastatic outgrowth. In human breast cancer, the frequency of CD39$^+$PD-1$^+$CD8$^+$ but not total CD8$^+$ T cells correlates with delayed metastatic relapse after resection (disease-free survival), thus underlining the biological relevance of CD39$^+$PD-1$^+$CD8$^+$ T cells for controlling experimental and human breast cancer. Thus, we suggest that a primary breast tumor could prime a systemic, CD39$^+$PD-1$^+$CD8$^+$ T cell response that favors metastatic dormancy in the lungs.

[1] Institute of Experimental Immunology, University of Zurich, Zurich, Switzerland. [2] Comprehensive Cancer Center Zurich, Zurich, Switzerland. [3] Department of Pathology and Molecular Pathology, University Hospital Zurich, Zurich, Switzerland. [4] Department of Orthopaedics, Balgrist University Hospital, University of Zurich, Zurich, Switzerland. [5] Department of Immunology, Juntendo University School of Medicine, Tokyo, Japan. [6] Present address: Department of Medicine, Mount Sinai St. Luke's & Mount Sinai West, Icahn School of Medicine at Mount Sinai, New York, NY, USA. [7] Present address: Cutiss AG, Schlieren, Switzerland. [8] Present address: Department of Hematology and Oncology, University Hospital Zurich, Zurich, Switzerland. [9] These authors contributed equally: Paulino Tallón de Lara, Héctor Castañón. ✉email: vandenbroek@immunology.uzh.ch

Metastasis is the major cause of death in patients with breast cancer. The metastatic behavior of this disease is heterogeneous regarding affected organs as well as timing[1,2]. Whereas some patients develop metastasis shortly after or even before diagnosis, others show metastatic lesions only years or decades after removal of the primary tumor[1,3]. In fact, ~20% of disease-free patients will relapse within 7 years after resection, and patients that appear to be cured from breast cancer have a higher mortality than the rest of the population even 20 years after surgery[4,5]. Such late relapses are thought to result from disseminated cancer cells (DCCs) that reached different organs but remained dormant for several years[6]. Indeed, the presence of DCCs in the bone marrow of breast cancer patients loosely correlates with the development of metastatic relapse[7,8].

The cancer cell-intrinsic and -extrinsic mechanisms governing metastatic dormancy are poorly understood. Some cancer cell-intrinsic factors have been associated with dormancy, such as inhibition of the PI3K-AKT pathway[9], activation of p38 or triggering of an endoplasmic reticulum stress response[10]. In addition, the microenvironment in the (pre)metastatic organ may induce dormancy[11] by different signals such as an endothelial cell-derived thrombospondin-1[12] or TGF-β2 produced in the bone marrow[13]. Only few studies addressed the influence of cancer cell-extrinsic factors, such as innate[14,15] and adaptive immunity[16,17] on dormancy. Although infiltration of the primary tumor by T cells was shown to correlate with a good prognosis[18], it remains unclear whether this correlation is explained by cytotoxic T cells that eliminate cancer cells or by T cells that prevent the outgrowth of cancer cells and induce dormancy[19,20].

Here we sought to understand the mechanism that governs metastatic dormancy of DCCs and discovered that the primary tumor primes a systemic, CD8+ T cell response that prevents metastatic outgrowth of DCCs in the lungs. The protective T cells expressed markers that are generally associated with activation and exhaustion. Promoting such a response may provide a rationale for the development of future immunotherapies that aim to prevent metastatic relapse.

Here, we show that CD39+PD-1+CD8+ T cells mediates metastatic dormancy in a preclinical model for breast cancer and correlate with increased disease-free survival post-resection in breast cancer patients.

## Results

**Disseminated 4T07 breast cancer cells are dormant**. We used two preclinical models for breast cancer that were originally derived from the same spontaneous tumor in BALB/c mice, 4T1 and 4T07[21] (Fig. 1a). 4T1 orthotopic breast cancer produces macro-metastasis in different organs[22,23], whereas 4T07 is essentially non-metastatic. Some studies, however, showed the presence of disseminated 4T07 cells in distal organs[21,24].

Orthotopically injected 4T1 and 4T07 cells showed comparable breast cancer growth and incidence in immunocompetent BALB/c mice, although 4T1 tumors grew faster at late stages (Supplementary Fig. 1a and b). But, whereas 4T1 breast cancer readily induced aggressively growing and macroscopically visible lung metastasis, 4T07 did not (Supplementary Fig. 1c and d). Despite absence of macro-metastasis, we found disseminated 4T07 cells in the lungs (Supplementary Fig. 1e) of all mice, suggesting that 4T07 DCCs seed the lungs but fail to grow out. Presence of dormant DCCs in the lungs of all mice bearing a 4T07-mCh primary tumor was confirmed and quantified by immunofluorescence (Fig. 1b–d, and Supplementary Fig. 1f). Disseminated 4T07 cells were present as single cells and were Ki67-negative, as has been previously described for dormant DCCs[15,17]. The fact that we found non-proliferating, disseminated 4T07 21 days after resection of the primary tumor (Fig. 1e–g) further underscores the dormant state of 4T07 DCCs.

**Breast cancer induces CD8+ T cells resulting in dormancy**. We first investigated whether cancer cell-intrinsic features influenced the outgrowth of DCCs in the lungs by injecting 4T1 or 4T07 breast cancer cells intravenously (i.v.) into naïve mice (Fig. 2a). Under these experimental conditions, both cell lines comparably formed macro-metastases (Fig. 2b), suggesting that both cell lines can progressively grow in the lungs.

The myeloid cell compartment was shown to undergo cancer-induced changes that promote metastasis by creating a pre-metastatic niche in the 4T1 model[25,26]. Specifically, 4T1 primary tumors induce systemic neutrophilia[27] and splenomegaly[26], as well as accumulation of inflammatory monocytes[28], eosinophils, neutrophils[27] and alveolar macrophages[29] in the pre-metastatic lung. We therefore compared 4T1- and 4T07-induced changes on the myeloid compartment in blood and lungs and observed no differences regarding splenomegaly (Supplementary Fig. 2a), or neutrophilia (Supplementary Fig. 2b). The lungs of mice with 4T1 or 4T07 breast cancer were similar concerning the presence of inflammatory monocytes (Supplementary Fig. 2c), neutrophils (Supplementary Fig. 2d), eosinophils (Supplementary Fig. 2e) and alveolar macrophages (Supplementary Fig. 2f) as determined by flow cytometry (Supplementary Fig. 2g). We saw the previously reported accumulation of myeloid cells in the lungs during breast cancer progression[27] in 4T1 and 4T07 alike (Supplementary Fig. 2). Thus, the difference in metastatic behavior of 4T1 and 4T07 breast cancer cannot be explained by systemic changes in the myeloid compartment.

Orthotopic 4T07 tumors induced progressively growing lung metastases in T cell-deficient BALB/c *Foxn1nu/nu* mice, whereas T cell-deficiency hardly influenced metastatic behavior of 4T1 tumors (Fig. 2c–e). To compare lung metastases in both strains, we analyzed wild type and *Foxn1nu/nu* mice at the same time point after injection and added an additional group of wild type mice in which breast tumors were allowed to progress until they matched the size in *Foxn1nu/nu* mice (WT late) (Fig. 2d and e). Thus, metastatic dormancy of disseminated 4T07 breast cancer cells completely depends on T cells. The fact that 4T1 cells are intrinsically more metastatic than 4T07 cells in immunodeficient mice reflects (a combination of) traits that are different between 4T1 and 4T07 cells, some of which are unrelated to T-cells. In fact, we think that the many differences between 4T1 and 4T07 cells preclude appropriate and conclusive comparison in vivo.

To study whether CD8+ T cells are responsible for metastatic dormancy, we depleted CD8+ T cells from mice followed by orthotopic injection of 4T07 cells and subsequent analysis of lung metastatic load by IVIS (Fig. 2f). While the growth of primary tumors was unaffected by CD8-depletion, disseminated 4T07-LZ cells grew out to macro-metastases in the absence of CD8+ T cells (Fig. 2g and h), suggesting that primary 4T07 breast cancer induces CD8+ T cell-dependent immunity.

To test the hypothesis that CD8+ T cells are essential for metastatic dormancy, we orthotopically injected untagged 4T07 cells (or PBS as control) followed by an i.v. challenge with luciferase-tagged 4T07-LZ cells 11 days later (Fig. 2i). If the primary tumor had induced protective immunity, we would expect a reduction of lung metastatic load. Because we measured lung metastatic load by bioluminescence, we specifically quantified the i.v. injected, luciferase-tagged 4T07 cells. The presence of a primary 4T07 breast tumor prevented metastatic outgrowth of i.v. injected 4T07-LZ cells (Fig. 2i–k) but did not influence the amount of seeding as measured 0.5 and 3 h after i.v. injection

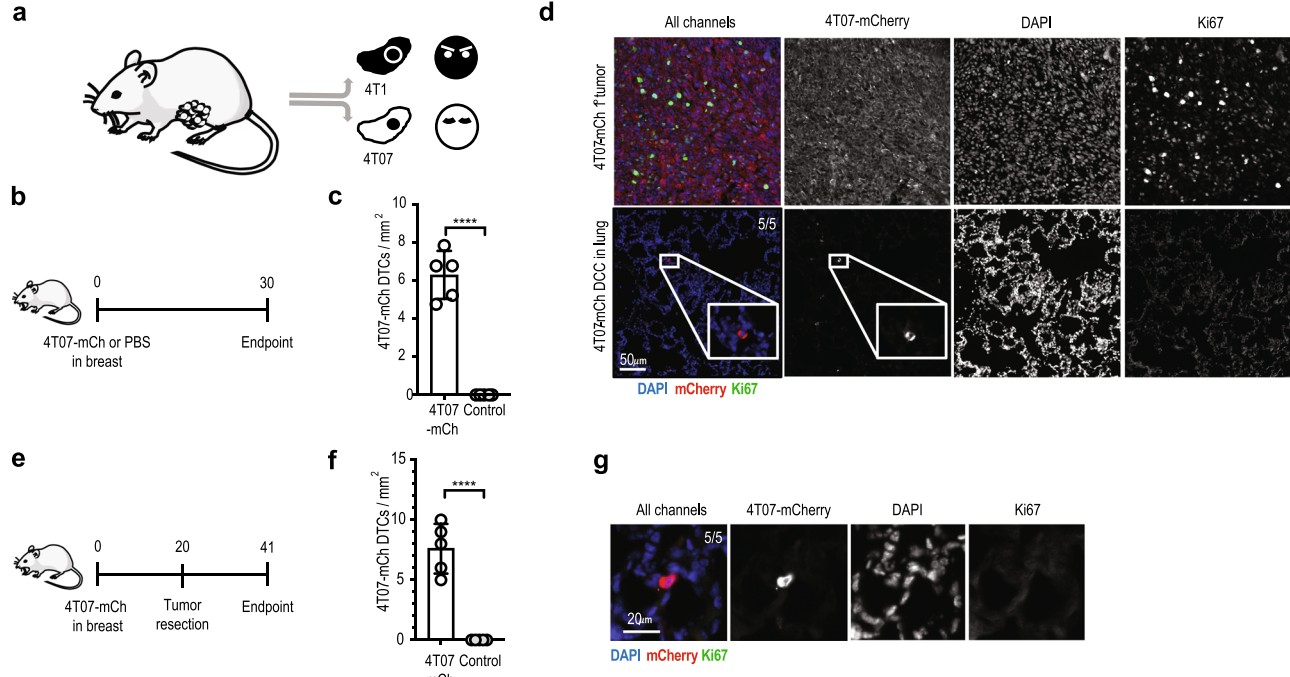

**Fig. 1 4T07 but not 4T1 breast cancer cells are dormant after dissemination to the lungs. a** Diagram of the origin of 4T1 and 4T07 cell lines[21].
**b** Experimental design. 4T07-mCh cells ($10^5$) or PBS were injected into the mammary fat pad of female BALB/c mice. Analysis was performed 30 d later.
**c** Quantification of disseminated 4T07-mCh cells by quantitative pathology, ****$p < 0.0001$. **d** 4T07-mCh cells ($10^5$) were injected into the mammary fat
pad of female BALB/c mice. Analysis was performed 35 d later. Representative section of the primary tumor (upper panels) and lung (lower panels) at the
endpoint. Disseminated 4T07-mCh cells were detected as single, Ki67-negative cells in the lungs of 5 out of 5 mice. Disseminated 4T07-mCh cells are
shown in red, Ki67 in green, DAPI in blue. Scale bar indicates 50 µm. **e** Experimental design. 4T07-mCh cells ($10^5$) were injected into the mammary fat pad
of female BALB/c mice. The tumor was resected 20 d after injection and analysis was performed 21 d after resection (endpoint). **f** Quantification of
disseminated 4T07-mCh cells by quantitative pathology. ****$p < 0.0001$. **g** Representative section of the lung at the endpoint. Disseminated 4T07-mCh
cells were detected as single, Ki67-negative cells in the lungs of 5 out of 5 mice. Disseminated 4T07-mCh cells are shown in red, Ki67 in green, DAPI in
blue. Scale bar indicates 20 µm. Each symbol represents an individual mouse. Five mice per group. 2-tailed Student's $t$-test. The bar represents the mean ±
SD. Results are representative of three independent experiments.

(Supplementary Fig. 3a and b). At the endpoint, i.v. injected cells
were present as disseminated, non-cycling single 4T07 cells in the
lungs (Supplementary Fig. 3c and d). Resection of the primary
tumor before i.v. challenge did not interfere with dormancy.
Specifically, i.v. injected 4T07-mCh cells readily induced macro-
scopically visible lung metastasis in control mice (PBS in breast
and mock surgery; Supplementary Fig. 3e, f), whereas only single,
non-proliferating 4T07 cells were detected in the lungs of 4T07
tumor-bearing mice, independent of resection (Supplementary
Fig. 3e, g). We confirmed these results in a similar experimental
set-up using 4T07-LZ cells and bioluminescence as read out
(Supplementary Fig. 4a, b). Depletion of CD8$^+$ cells before i.v.
challenge enabled metastatic outgrowth (Fig. 2l and m),
confirming the dependence of dormancy on CD8$^+$ T cells. In
contrast, 4T07 nor 4T1 breast cancer protected against experi-
mental 4T1 lung metastasis, although 4T07-induced systemic
immunity reduced the metastatic load resulting from i.v. injected
4T1-LZ cells (Supplementary Fig. 4c and d). Although the lung
metastatic load resulting from i.v. injected 4T1 cells is reduced in
4T07-bearing mice, the lesions are still progressive (i.e., multi-
cellular and Ki67$^+$) and not dormant (Supplementary Fig. 4a–d).
Furthermore, we think that the number of adoptively transferred
CD39$^+$PD-1$^+$CD8$^+$ T-cells is simply insufficient to control
4T1 cells that have an intrinsically higher metastatic potential.

Together, these data suggest that 4T07 breast cancer induces a
systemic, protective immune response that mediates dormancy of
DCCs in the lungs.

**CD39$^+$PD-1$^+$CD8$^+$ T cells mediate metastatic dormancy.** First,
we analyzed the presence of different immune cells in primary
4T1 and 4T07 tumors (Supplementary Fig. 5a). We found no
differences in the number of eosinophils (Supplementary Fig. 5b),
neutrophils (Supplementary Fig. 5c), NK cells (Supplementary
Fig. 5d) or inflammatory monocytes (Supplementary Fig. 5e), but
4T1 tumors contained a higher number of macrophages than
4T07 tumors (Supplementary Fig. 5f).

Since we found that 4T07 metastatic dormancy depends on
CD8$^+$ T cells, we characterized these cells in 4T1 and 4T07
primary tumors by high-dimensional flow cytometry. First, we
gated on T cells (Supplementary Fig. 6a), visualized all the
markers in our 23-parameter panel using a two-dimensional $t$-stochastic
neighbor embedding (tSNE) projection[30] (Fig. 3a) and clustered
the cells using FlowSOM[31,32] (Fig. 3a and b, upper panels and
lower left panel). Based on the median marker intensities observed
in the clusters (Fig. 3b, lower right panel), we annotated the major
cell populations (CD8$^+$ T cells, CD4$^+$ T cells, and CD4$^+$ CD25$^+$
regulatory T cells) and determined their frequency and the
position of each population in the tSNE projection. In addition,
not only the number of CD39$^+$PD-1$^+$CD8$^+$ T cells was higher in
4T07 than in 4T1 tumors, but also their proportion of CD8$^+$
T cells. The frequency of regulatory T cells was similar in 4T1 and
4T07 tumors, while conventional CD4$^+$ cells were more abundant
in 4T1 tumors compared to 4T07. In contrast, 4T07 primary
tumors contained a higher proportion of CD8$^+$ T cells (Fig. 3b,
lower left panel).

 3

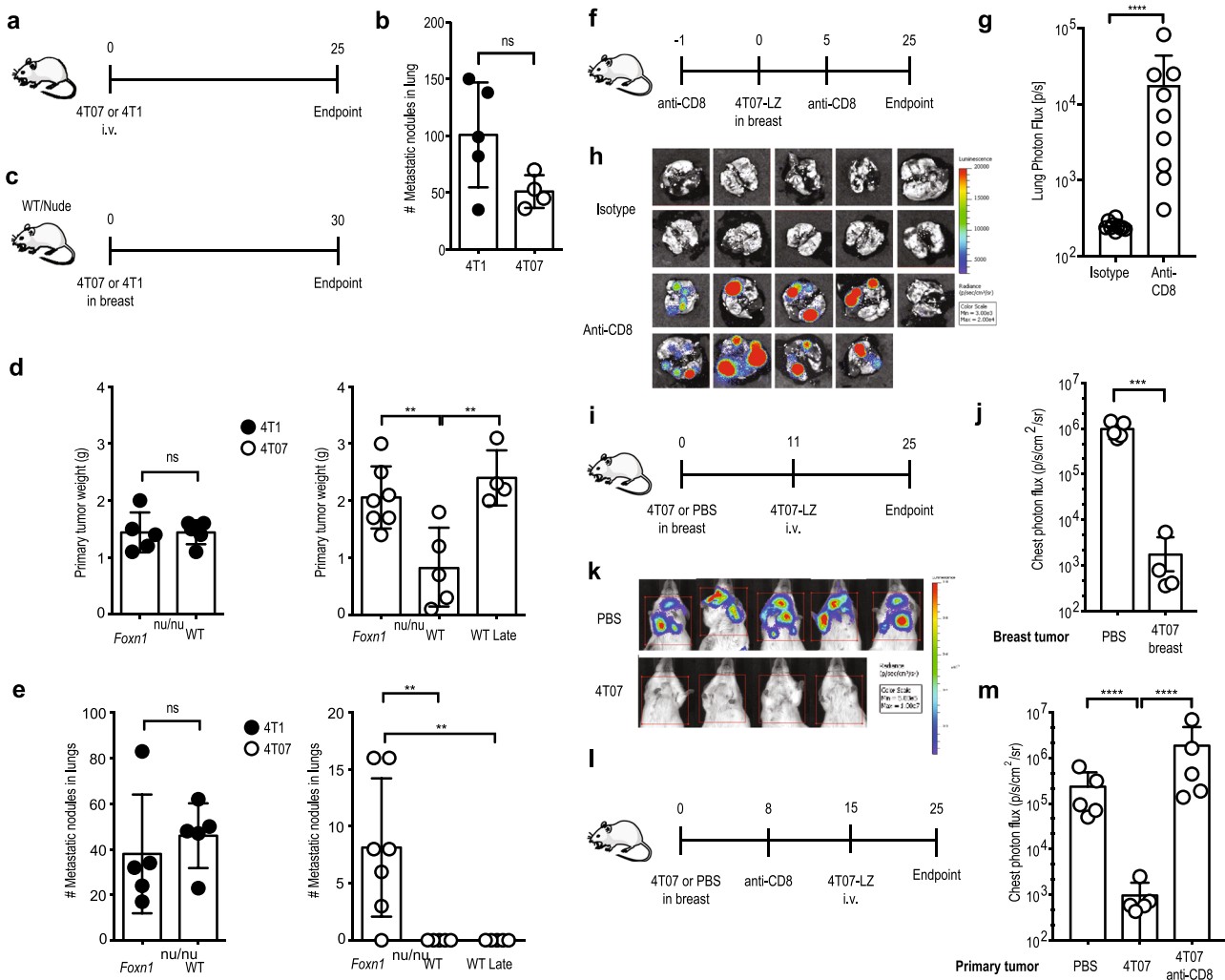

**Fig. 2 Induction of protective immunity by the primary tumor mediates metastatic dormancy. a** Experimental design. 4T1 or 4T07 cells ($3 \times 10^5$) were injected i.v. into female BALB/c mice. Analysis 25 d later. **b** Number of lung metastatic nodules. 4T1, $n = 5$; 4T07, $n = 4$. **c** Experimental design. 4T1 or 4T07 cells ($10^5$) were injected into the mammary fat pad of female BALB/c or BALB/c $Foxn1^{nu/nu}$ mice. Analysis 30 d later. Lungs of "WT Late" mice were analyzed when the 4T07 tumor size reached the size of the BALB/c $Foxn1^{nu/nu}$ group (d 35). **d** Weight of primary tumors. Left panel with closed symbols, 4T1. Both groups $n = 5$. Right panel with open symbols, 4T07. $Foxn1^{nu/nu}$, $n = 7$; WT, $n = 5$; WT late, $n = 4$, **$p = 0.006$ ($Foxn1^{nu/nu}$ vs. WT), **$p = 0.0027$ ($Foxn1^{nu/nu}$ vs WT Late). **e** Number of lung metastatic nodules. Left panel with closed symbols, 4T1; right panel with open symbols, 4T07, **$p = 0.071$ for both comparisons. **f** Experimental design. Female BALB/c mice received 0.5 mg anti-CD8 or isotype control i.p. at days $-1$ and $+5$ relative to injection of $10^5$ 4T07-mCh cells in the mammary fat pad. Analysis 25 d later. **g** Lung metastatic load measured by bioluminescence, ****$p < 0.0001$. Anti-CD8, $n = 9$; isotype $n = 10$. **h** Bioluminescence images. Anti-CD8, $n = 9$; isotype $n = 10$. **i** Experimental design. 4T07 cells ($10^5$) or PBS were injected into the mammary fat pad of female BALB/c mice on d 0. On d 11, $3 \times 10^5$ 4T07-LZ cells were injected i.v.; analysis of lung metastatic load on d 25. **j** Quantification of lung metastatic load by bioluminescence. 4T1, $n = 5$; 4T07, $n = 4$, ***$p = 0.0021$. **k** Bioluminescence images. **l** 4T07 cells ($10^5$) or PBS were injected into the mammary fat pad of female BALB/c mice on d 0. On day 8, mice were injected i.p. with 500 µg anti-CD8 or isotype control. On d 15, $3 \times 10^5$ 4T07-LZ cells were i.v. injected and lung metastatic load was quantified on d 25. **m** Quantification of the lung metastatic load by bioluminescence on d 25. All groups $n = 5$, ****$p < 0.0001$. Each symbol represents an individual mouse. ns = not significant. 2-tailed Student's $t$-test with Welch's correction (**b**, **d** left panel, **e** left panel, **g**, **j**); 2-sided ANOVA with Bonferroni correction (**d** right panel, **e** right panel, **m**). The bar represents the mean ± SD. Results are representative of 3 (**f**, **g**, **h**) or 2 (all other) independent experiments.

Of note, within the CD8 compartment (Fig. 3c), we observed that 4T07 breast cancer accumulated CD39+PD-1+CD8+ T cells (Fig. 3d).

To substantiate that enrichment of CD39+PD-1+CD8+ T cells is the key immunological difference between 4T07 and 4T1 breast cancers, we analyzed our high-dimensional flow cytometry data using the unbiased representation-learning algorithm CellCnn[33,34]. In agreement with FlowSOM analysis, CellCnn detected a population with high expression of CD39, PD-1, LAG3, and Tim-3 (Fig. 3e and Supplementary Fig. 6a and b). Indeed, the two markers that best defined the population and showed the biggest differential abundance

in terms of the Kolmogorov–Smirnov two-sample test between the whole-cell population and the selected cell subsets (higher KS value) were PD-1 and CD39 (Fig. 3e). In addition, CD39+PD-1+CD8+ T cells expressed more of LAG-3 and Tim-3 (Fig. 3f), which characterizes this population as effector cells that experienced recent T cell receptor engagement[35,36] but may also indicate exhaustion[37,38]. This population was around three times more abundant in 4T07 compared to 4T1 tumors (Fig. 3g and h, $p < 0.001$). Therefore, we analyzed the functionality of CD39+PD-1+CD8+ T cells sorted from 4T07 tumors and confirmed the production of IFNγ and TNFα (Supplementary Fig. 6c). In addition, CD39+PD-1+CD8+ T cells

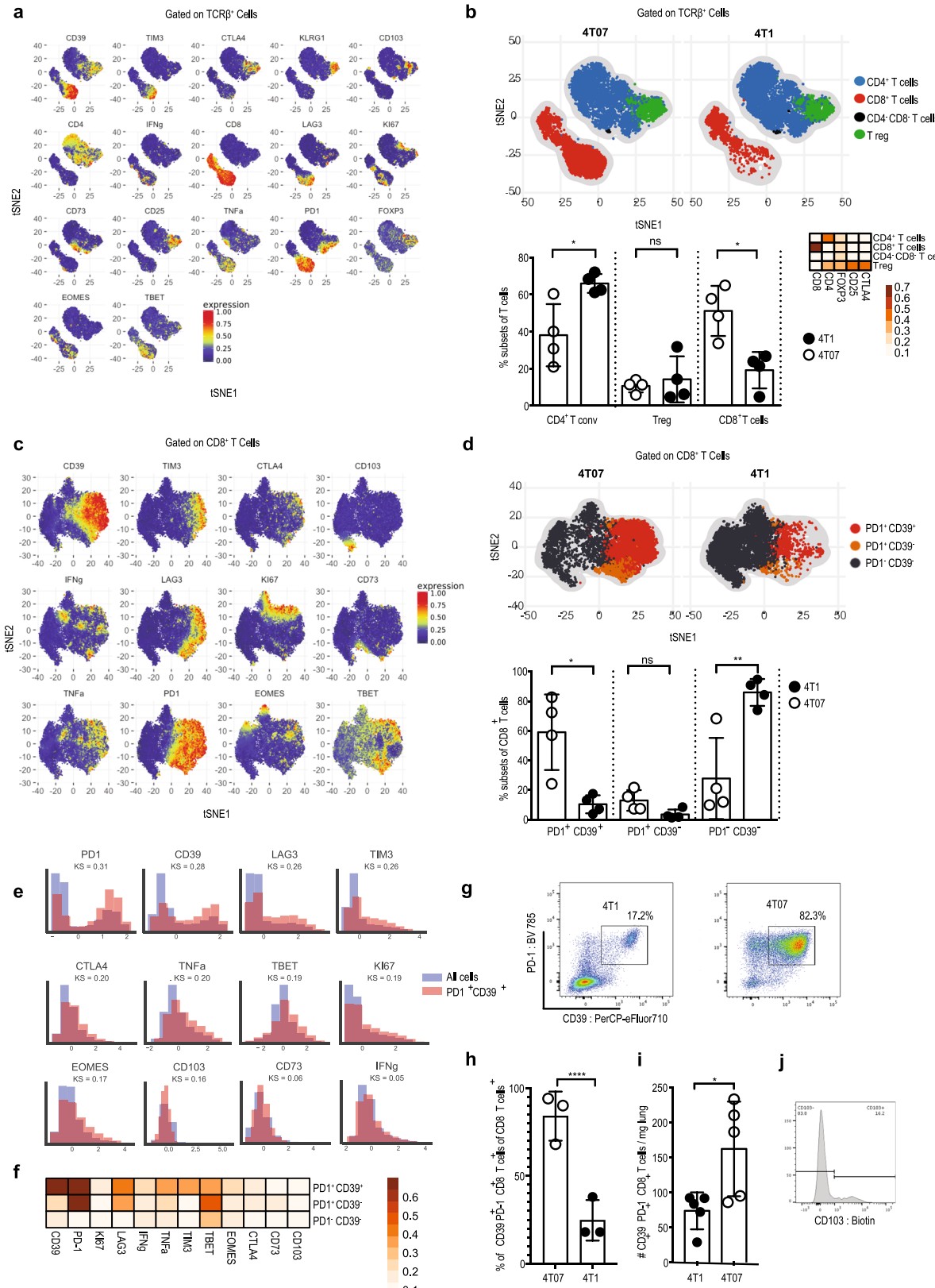

displayed cytotoxicity toward 4T07 as well as 4T1 cells in vitro (Supplementary Fig. 6d). We found increased presence of such CD39+PD-1+CD8+ T cells already at d 10 in primary 4T07 (and not 4T1) breast cancers (Supplementary Fig. 7a–d). Together, these results suggest that CD39+PD-1+CD8+ T cells in 4T07 tumors possess protective effector functions.

Importantly, CD39+PD-1+CD8+ T cells are found in the lungs of mice with primary 4T07 breast cancer (Fig. 3i). A proportion of this population expresses CD103, hence resembling tissue-resident memory T cells[39,40] (Fig. 3j). Our observation that CD39+PD-1+CD8+ T cells did not express CD103 in the primary tumor (Fig. 3f) is in line with recent data showing that

**Fig. 3 CD39+PD-1+CD8+ T cells emerge in dormant breast cancer.** Female BALB/c mice were injected with $10^5$ 4T1 or 4T07 cells in the mammary fat pad and primary tumors were analyzed by flow cytometry 20 d later. **a** t-SNE visualization of markers after gating on single, live, CD45+ TCRβ+ CD44+ cells. **b** Upper panel: t-SNE plots of the main T cell subsets. Lower left panel: frequency of the main T cell subsets in both groups. Each symbol represents an individual mouse, *p = 0.019 (CD4+ Tconv), *p = 0.011 (CD8+ T cells). Lower right panel: Heat map with marker expression of the main T cell subsets. **c** t-SNE visualization of markers on CD8+ TCRβ+ cells. **d** Upper panel: t-SNE plots of the main CD8+ T cell subsets identified by FlowSOM algorithm. Lower panel: frequency of the main CD8+ T subsets identified by FlowSOM algorithm in both groups, *p = 0.028, **p = 0.0069. **e** Scaled histograms of arcsinh-transformed marker expression showing the relative marker distribution of the population identified by CellCnn (red) among all CD8+ T cells (blue). KS indicates the Kolmogorov–Smirnov two-sample test between the whole-cell population and the selected cell subsets. **f** Heat map with marker expression of the main CD8+ T cell subsets identified by FlowSOM algorithm. **g** Representative FACS plot of 4T1 and 4T07 primary tumors after gating on CD8+ TCRβ+. **h** Frequency of the population of total CD8+ identified by CellCnn in both models, ****p < 0.0001. **i** Number of CD39+PD-1+CD8+ T cells in the lungs, *p = 0.0317. **j** Representative CD103-staining of live CD39+PD-1+CD8+ T cells from the lungs. Each symbol represents an individual mouse. n = 4 per group for all panels, except panel h (n = 3) and i (n = 5). Two-tailed Student's t-test with Welch's correction. The bar represents the mean ± SD.

tissue-resident-like memory cells in human breast cancer are CD103-negative[41]. Thus, accumulation of CD39+PD-1+CD8+ T cells is the main immunological difference between dormant and metastatic tumors.

Because the CD8+ cells specifically emerging in 4T07 express PD-1, we wondered whether blocking PD-1 may improve their presumed protective effect. Indeed, immunotherapy with anti-PD-1 reduce the number of disseminated 4T07 cells in the lungs approximately twofold (Fig. 4a and b), suggesting that the PD-1+ subset of CD8+ T cells comprises protective effector cells. To investigate whether the CD39+PD-1+CD8+ T cell subset indeed controls metastatic outgrowth and therefore mediates metastatic dormancy, we sorted this population from 4T07 breast cancer and adoptively transferred them to naïve BALB/c mice followed by an i.v. injection of 4T07-LZ cells (Fig. 4c and d). Adoptive transfer of CD39+PD-1+CD8+ T cells prevented 4T07 lung metastasis, whereas injection of PBS or CD8+ T cells lacking CD39+PD-1+ cells (referred to as other CD8+ T cells) did not (Fig. 4e and f). In these experiments, we did not visualize the dormant cells, thus leaving some room for the possibility that the i.v. injected cells actually never reached the lungs when CD39+PD-1+CD8+ T-cells were transferred. For this reason, we performed the same experiment and visualized disseminated 4T07-mCh cells in the lungs. We observed metastatic outgrowth in all lungs of mice that did not receive CD39+PD-1+CD8+ T-cells; in contrast, in mice that received CD39+PD-1+CD8+ T-cells, we found single, non-proliferating and thus dormant 4T07-mCh cells in the lungs (Supplementary Fig. 8). Thus, our results suggest that CD39+PD-1+CD8+ T cells are necessary and sufficient to control disseminated 4T07 breast cancer cells.

The presence of non-proliferating disseminated 4T07 cells in the lungs suggests that CD8+ T cells mediate dormancy by cell cycle arrest. Therefore, we tested whether CD39+PD-1+CD8+ T cell-derived IFNγ and TNFα can induce senescence[42] by exposing 4T07 cells to those cytokines in vitro. We observed a significant increase in the number of senescent cells as measured by expression of senescence-associated β-galactosidase activity (Supplementary Fig. 9a and b). To confirm the relevance of IFNα/TNFα-induced senescence induction in vivo, we blocked both cytokines in mice with 4T07-mCh breast cancer and analyzed the status of disseminated 4T07 cells in the lungs by immunofluorescence (Fig. 4g). Blocking IFNγ and TNFα allowed the metastatic outbreak of disseminated cells as illustrated by the presence of clusters of proliferating 4T07 cells (Fig. 4h). We observed such metastatic outbreaks in the lungs of 4 out of 8 mice that were treated with anti-IFNγ and anti-TNFα. In the lungs of isotype-treated mice, we found no metastatic outbreaks; instead, we observed single, non-proliferating disseminated cells in the lungs of all mice analyzed. Thus, the mere presence of multicellular metastatic 4T07 nodules in the lungs after blocking IFNγ and TNFα shows that induction of dormancy depends on these two cytokines.

Thus, 4T07 primary tumors prime systemic CD8+ T cells that induce cell cycle arrest of DCCs by IFNγ and TNFα and thereby prevent overt metastatic disease.

**Effector function and exhaustion in CD39+PD-1+CD8+ T cells.** To understand why this population controls DCCs, we sorted CD39+PD-1+CD8+ and other CD8+ T cells from 4T07 breast cancer and compared their transcriptome (Fig. 5a and b and Supplementary Fig. 10a). CD39+PD-1+CD8+ cells show over-representation of transcripts associated with effector function including *Prf1, Fasl, Gzmf, Gzme, Gzmd, Gzmc, Gzmb, Gzmk*, and *Ifng* (Fig. 5c). This was confirmed by gene set enrichment analysis (GSEA)[43] showing that the transcriptome of the sorted CD39+PD-1+CD8+ cells shows strong similarities with human tissue-resident CD8+ T cells (Fig. 5d), which were recently shown to correlate with improved survival in triple negative breast cancer[19]. This is in line with their unique capacity cells to prevent metastatic progression. Furthermore, the fact that dormancy is never observed in 4T1 breast cancer is explained by the low number of such protective T cells in 4T1-bearing hosts (Fig. 3) and not by their transcriptional signature, since CD39+PD-1+CD8+ cells from 4T1 and 4T07 tumors did not show many differences (Supplementary Fig. 10b–d). The transcriptome of CD39+PD-1+CD8+ cells was also enriched in markers of activation and exhaustion such as *Vsir* (VISTA), *Tnfrsf18* (GITR), *Tigit, Cd224* (2B4), *Tnfrsf9* (4-1BB), *Tnfrsf4* (OX-40), *Icos, Lag3, Ctla4*, and *Havcr2* (Tim-3). This is consistent with their overexpression of the transcription factor TOX (Fig. 5c) which leads to expression of exhaustion markers and persistence of effector function during chronic antigen stimulation[44].

Taken together, CD39+PD-1+CD8+ cells have a unique transcriptomic signature defined by the co-expression of immune checkpoint molecules and effector proteins.

**CD39+PD-1+CD8+ T cells correlate with survival in human breast cancer.** To investigate the clinical relevance of our pre-clinical findings, we analyzed primary breast cancer tissues from a cohort of 54 patients, who had a metastatic relapse after primary tumor resection (Supplementary Table 1) for the presence of CD8, CD39, PD-1, and epithelial cells using 5-plex immunofluorescence (Fig. 6a). Our cohort shows the typical survival curves for the individual subtypes, suggesting that it is representative (Supplementary Fig. 11a). Patients with a high density of intra-tumoral CD39+PD-1+CD8+ T cells had a significantly longer disease-free survival after surgery than patients with a low density of such cells (Fig. 6b). The density of extra-tumoral CD39+PD-1+CD8+ T cells did not correlate with disease-free survival (Supplementary Fig. 11b), whereas to density of CD39+PD-1+CD8+ T cells independent of their location did (Supplementary Fig. 11c). In addition, we observed similar data when focusing on luminal A and B patients (Supplementary Fig. 11d and e). The density of

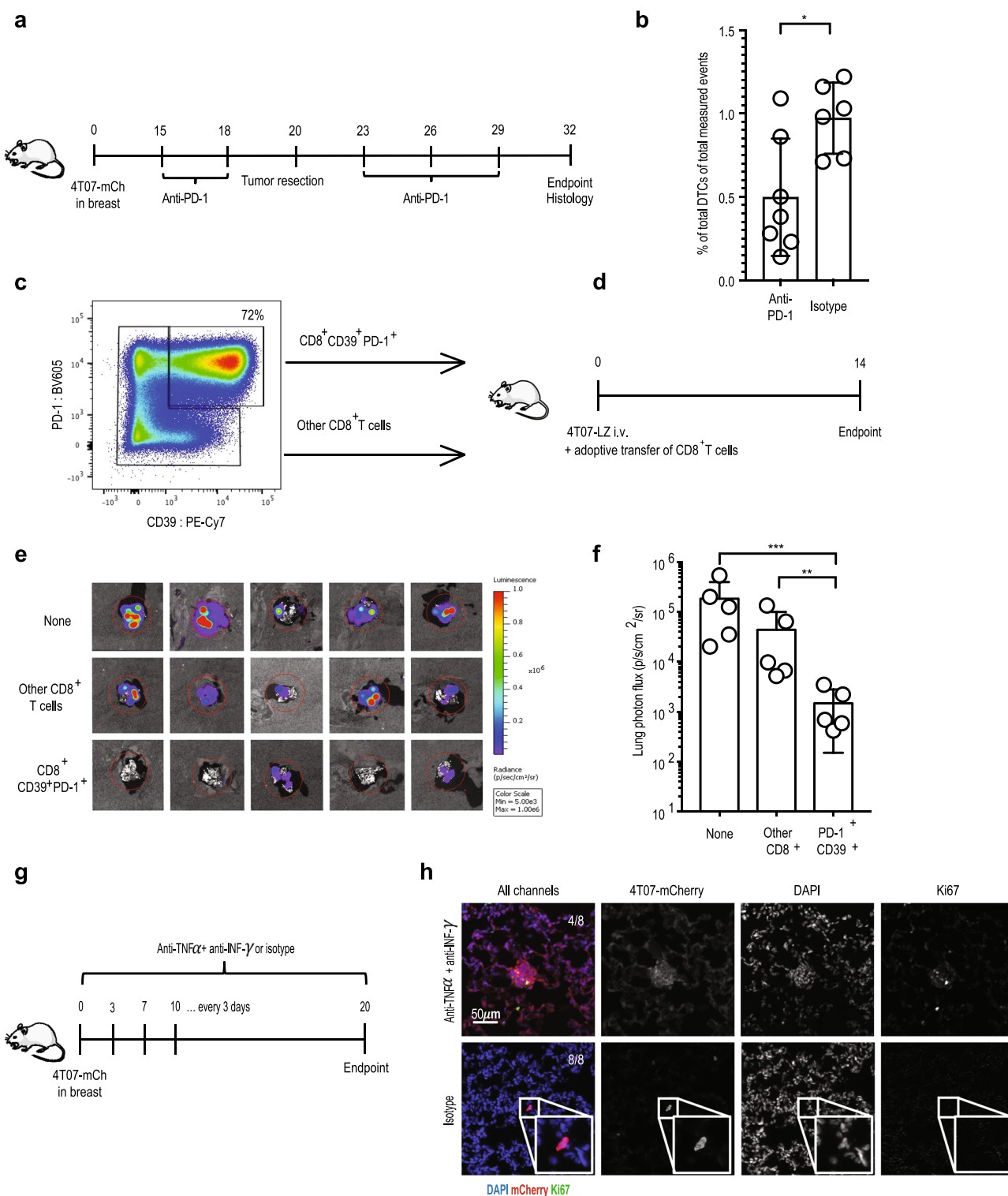

intra-tumoral CD39$^+$PD-1$^+$CD8$^+$ T cells was not an independent variable according to multivariate Cox regression analysis (Supplementary Table 1). Importantly, the density of intra-tumoral CD8$^+$ T cells showed no correlation with disease-free survival (Fig. 6c), strongly supporting our preclinical observations, namely that that disseminated cancer cells are controlled by this specific subset of CD39$^+$PD-1$^+$CD8$^+$ T cells.

## Discussion

We discovered that breast cancer can prime a systemic, CD8$^+$ T cell response that mediates metastatic dormancy in the lungs and consequently, protects against clinical metastatic outgrowth. We thus describe an unexpected trait of primary tumors, in addition to their well-known metastasis-promoting capacity[25,26,45,46]. Using high-dimensional single-cell profiling of primary tumors,

**Fig. 4 Tumor-associated CD39+PD-1+CD8+ T cells prevent metastatic outgrowth. a** Experimental design. 4T07-mCh cells ($10^5$) were injected into the mammary fat pad of female BALB/c mice. The breast tumor was resected on d 20 and histological analysis of lungs was performed on d 32. Mice received an i.p. injection with 250 µg anti-PD-1 or isotype control on d 15, 18, 23, 26, and 29. **b** Enumeration of disseminated 4T07 cells by quantitative pathology (right panel). Each symbol represents an individual mouse. Isotype, $n = 6$; anti-PD-1, $n = 7$, $*p = 0.0152$, (two-tailed Student's $t$-test with Welch's correction). The bar represents the mean ± SD. **c** Gating used for sorting of CD44+CD39+PD-1+CD8+ and the rest of the CD44+CD8+ population (termed other CD8+ T cells) from established 4T07 orthotopic breast tumors. **d** Experimental design. Two-hundred-thousand sorted CD44+CD39+PD-1+CD8+ or other CD44+CD8+ T cells were transferred after injection of $10^5$ 4T07-LZ cells into female BALB/c mice. All injections were given intravenously. Lung metastatic load was determined by bioluminescence 14 d later. **e** Bioluminescence of lungs at endpoint. **f** Quantification of lung metastatic load by bioluminescence, $**p = 0.0058$, $***p = 0.0002$. Each symbol represents an individual mouse. Five mice per group. $**p < 0.01$, $***p < 0.001$ (ANOVA with Bonferroni's correction). The bar represents the mean ± SD. **g** Experimental design. 4T07-mCh cells ($10^5$) were injected into the mammary fat pad of female BALB/c mice. Mice received an i.p. injection with 500 µg anti-IFNγ plus 500 µg anti-TNFα every 3rd day. Control mice received isotype control antibody. On d 20, 4T07-mCh cells were visualized in lungs by immunofluorescence. **h** Two representative examples showing clusters of proliferating 4T07-mCh cells in the lungs of mice treated with anti-IFNγ plus anti-TNFα. Proliferating 4T0-mCh7 cells were detected in the lungs from 4 out of 8 mice.

we identified the protective CD8+ T cell population as PD-1+ and CD39+ cells, suggesting recent cognate interaction and tumor-specificity[20,47]. In line with our preclinical data, we found that a high density of intra-tumoral CD39+PD-1+CD8+ T cells significantly correlated with disease-free survival after resection in breast cancer patients. Even in patients with luminal A or B subtypes, where the tumor tissue is generally less infiltrated by CD8+ T cells[48], the presence of CD39+PD-1+CD8+ T cells correlated with disease-free survival after primary tumor resection. As in our preclinical model, the heterogeneous population containing all CD8+ T cells did not correlate with disease-free survival, strongly supporting the notion that CD39+PD-1+CD8+ T cells comprise a population of cells that is uniquely equipped to control disseminated cancer cells.

The role of CD39 in cancer is rather complex. On the one hand, there is evidence that CD39 expression marks tumor-specific T-cells and presence of such cells correlates with a better prognosis or immune-mediated control[47]. On the other hand, CD39 is constitutively expressed by some cancer cells and suppressive cell types in the tumor microenvironment (Tregs and myeloid cells) and catalyzes pro-inflammatory extracellular ATP, resulting in impaired CD8+ T cell-dependent tumor control[49,50]. The same papers described a reduction in tumor-associated macrophages and monocytes as well as a better T cell function (and tumor control) after blocking the enzymatic activity of CD39.

We observed that CD39+PD-1+CD8+ T cells express effector molecules but also markers that are typically associated with exhaustion. A similar population was recently described in human breast cancer[19,51–53]. Exhausted T cells are thought to have a reduced effector function and loss of proliferative capacity[54,55]. Nonetheless, exhausted T cells in breast cancer may be more functional than in melanoma[56].

In addition to local immunity, the induction of systemic immune responses by immunotherapy was shown to be essential for its efficacy[57], suggesting that protective CD8+ T cells are recruited to the tumor. Recirculating CD8+ T cells can also enter tissues, where they develop into CD103+ tissue-resident memory cells and may protect tissues against disseminating tumor cells by the induction of cell death or cell cycle arrest[58]. Indeed, we detected CD39+PD-1+CD8+ T cells in the primary tumor and in the lungs and found that a proportion expressed CD103 as a marker of tissue residency. It was recently shown that the presence of intra-tumoral CD103-expressing CD8+ T cells correlate with survival in human breast cancer[41].

As mentioned above, metastatic dormancy may result from two different states[59]: Disseminated single cells that are quiescent[60], or micro-metastases that remain stable through an equilibrium between proliferation and death[61]. We found that

disseminated 4T07 cells were present in the lungs as single non-cycling cells, suggesting dormancy through cell cycle arrest. Our observation that dormancy essentially depended on CD8+ T cells uncovers a novel mechanism underlying dormancy. We found that protective CD39+PD-1+CD8+ T cells produced TNFα and IFNγ, which were described to induce irreversible senescence in the context of Th1-mediated protection against cancer[42]; the same cytokines prevented proliferation of disseminated 4T07 breast cancer cells in the lungs.

It may be that environmental cues induce cancer stem cell-like properties in disseminated cancer cells[62], which is in line with the low cycling frequency. Such cancer stem cells would represent a potent reservoir for progressive metastasis[63] in response to environmental changes. Why dormant cancer cells sometimes awaken only after decades[3] is still enigmatic, and it cannot be excluded that different pathways converge to initiate growth of dormant lesions. Assuming CD8+ T cells are required for maintenance of dormancy it is conceivable that attrition of tissue-resident memory CD8+ T cells may result in awakening of dormant cells. Tumor resection and consequent loss of antigen may result in diminished immunological memory; tissue-resident memory cells are described to be particularly unstable in the absence of antigen[64,65]. Alternatively, dormant cells themselves may change, for example losing the expression of molecules preventing immune-mediated clearance[66]. Finally, environmental factors in the broadest sense may awaken dormant cancer cells. Such changes in the microenvironment include fibrosis[67], tissue remodeling[68], obesity[69], inflammation[70], disturbance of vascular homeostasis[71], glucocorticoids[72], or cigarette smoke[73]. How such factors induce cycling of dormant cells and whether immune cells can interfere at this stage is unknown.

Here we discovered that CD39+PD-1+CD8+ T cells mediate dormancy of disseminated cancer cells but are by themselves unable to fully eradicate all of the cancer cells. This is clinically highly relevant, as dormant cancer cells are in all likelihood the source of future metastatic disease. Because the factors determining awakening are not known and may even not be controllable, it is important to better understand the nature of dormant cancer cells and which pathways must be mobilized for their complete elimination.

## Methods

**Mice**. BALB/cJRj and BALB/cAnNRj-*Foxn1^nu/nu^* were purchased from Janvier labs (Roubaix, FR). NOD.Cg-*Prkdc^scid^Il2rg^tm1Wjl^*/SzJ mice were originally obtained from the Jackson Laboratory and provided by Christian Münz (University of Zurich, Switzerland). Mice were kept under specific pathogen-free conditions in individually ventilated cages at the Laboratory Animal Services Center at the University of Zurich. Mice had access to food and water ad libitum and were maintained on a 12-h light/dark cycle with environmental enrichment. All experiments were performed with 8–14-weeks-old female mice in accordance with

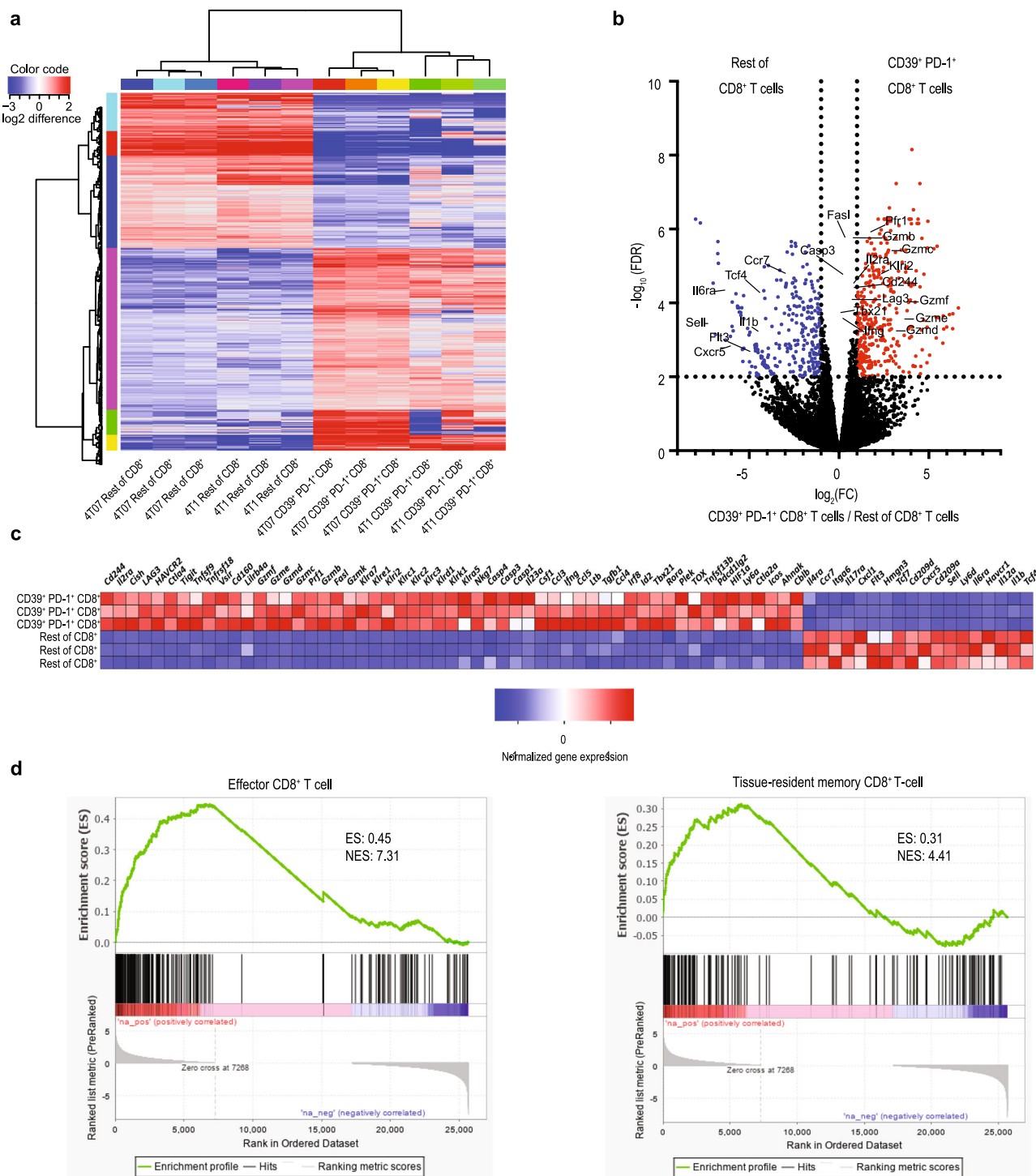

**Fig. 5 CD39⁺PD-1⁺CD8⁺ T cells have a unique transcriptional signature. a** Hierarchical clustering of significantly differentially expressed genes between sorted CD44⁺CD39⁺PD-1⁺CD8⁺ and CD44⁺CD8⁺ T cells. **b** Volcano plot comparing transcripts in CD44⁺CD39⁺PD-1⁺CD8⁺ T cells with those in other CD44⁺CD8⁺ T cells sorted from 4T07 breast tumors. The red symbols represent transcripts that are significantly over-expressed in CD44⁺CD39⁺PD-1⁺ CD8⁺ T cells, whereas the blue symbols represent significantly under-expressed transcripts. **c** Heat map showing relative expression of selected transcripts in CD44⁺CD39⁺PD-1⁺CD8⁺ (three top rows) and other CD44⁺CD8⁺ T cells (three bottom rows) identified by differential gene expression analysis. **d** Gene set enrichment analysis comparing the transcriptional profile to the data published by Goldrath et al.[80] (left panel) and Savas et al.[19] (right panel).

the Swiss federal and cantonal regulations on animal protection and were approved by The Cantonal Veterinary Office Zurich (156/2018).

**Cell lines**. 4T07 and 4T1 cells were a gift from Fred Miller (Karmanos Cancer Institute, Detroit, USA). Cells were cultured in Dulbecco's Modified Eagle's Medium (DMEM, Gibco) supplemented with 10% fetal bovine serum (FBS,

ThermoFisher Scientific), 2 mM L-glutamine and 2% penicillin/streptomycin (ThermoFisher Scientific). Cells were cultured at 37 °C in a humidified atmosphere with 5% $CO_2$. 4T1 and 4T07 were lentivirally transduced to express firefly luciferase and ZsGreen (pHIV-Luc-ZsGreen, addgene plasmid #39196) or mCherry (pCDH-CMV-mCherry-T2A-Puro, addgene plasmid #72264). Transduced cells were sorted based on expression of GFP or mCherry, respectively. Luciferase-ZsGreen is referred to as LZ, mCherry as mCh.

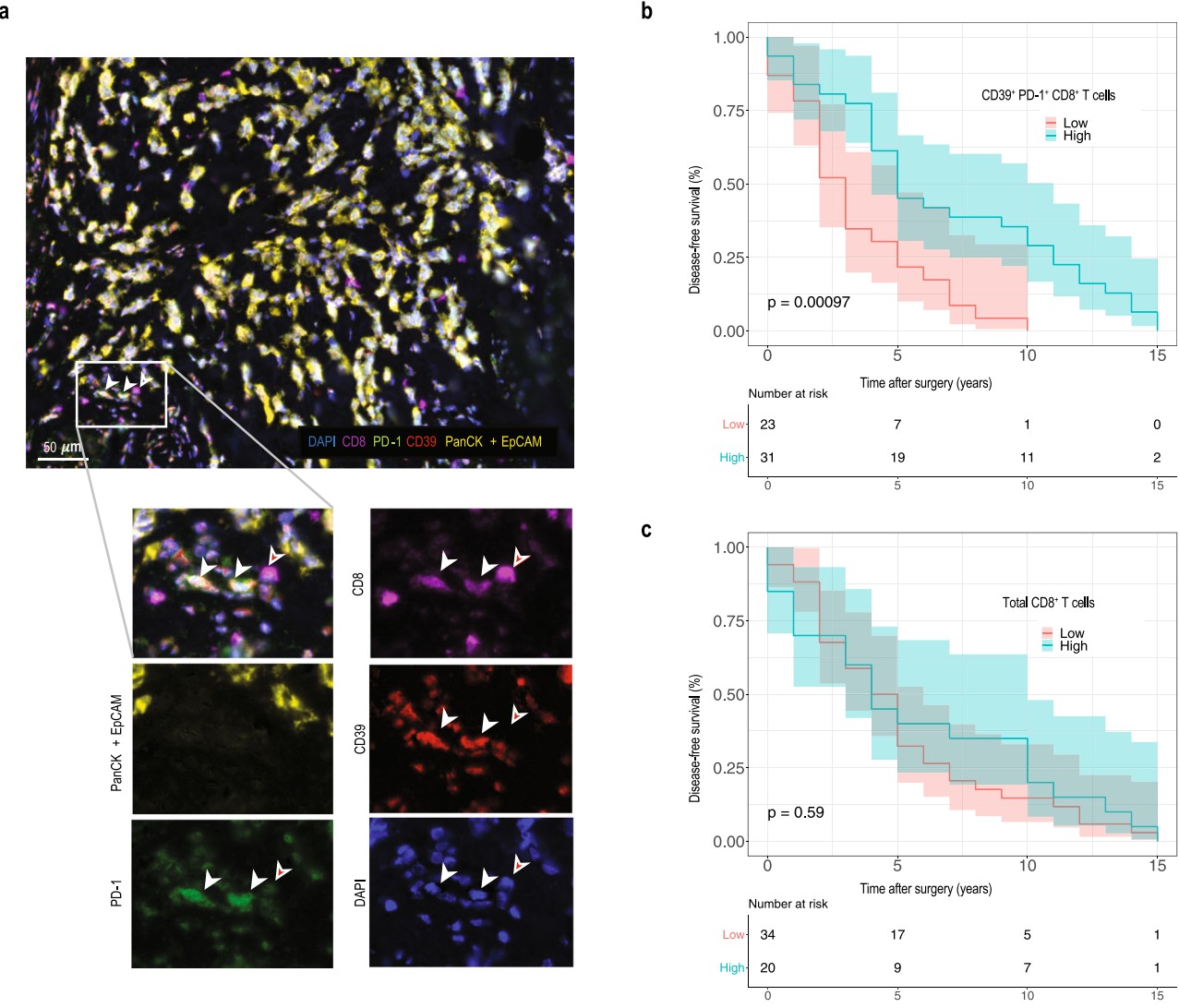

**Fig. 6 High density of intra-tumoral CD39⁺PD-1⁺CD8⁺ but not total CD8⁺ T cells correlates with disease-free survival in human breast cancer.**
**a** Representative images of 5-color multiplex immunofluorescence on human breast cancer. Staining shows epithelial cells (PanCK + EpCAM, yellow), CD8⁺ T cells (CD8, magenta), PD-1 (green), CD39 (red) and nuclear staining (DAPI, blue). Scale bar is 50 µm. **b** Disease-free survival of 54 patients with high or low number of intra-tumoral CD39⁺PD-1⁺CD8⁺ T cells. **c** Disease-free survival of 54 patients with high or low number of intra-tumoral CD8⁺ T cells. The threshold for separating patients with high and low CD39⁺PD-1⁺CD8⁺ T cell densities was defined using ROC curve analysis. Survival was compared for patients with high and low density of cells to be compared as indicated for the individual graphs by Kaplan–Meier curves. Confidence intervals are indicated as shaded areas surrounding survival curves. Significance was calculated by log-rank test. Patient numbers at risk are displayed for each 15 years of follow-up.

Only cells of early passages were used for experiments. Cells were regularly tested negative for mycoplasma by PCR analysis. Cells were also tested negative for 18 additional mouse pathogens by PCR (IMPACT II Test, IDEXX Bioanalytics).

**In vivo tumor experiments and treatments.** Hundred-thousand 4T1 or 4T07 cells in 50 µl PBS were injected into the fourth mammary fat pad. Alternatively, $3 \times 10^5$ cells in 50 µl PBS were injected into the lateral tail vein.

For resection of primary tumors, mice were anesthetized with 2.5% isoflurane and given 0.04 mg/kg fentanyl (Kantonsapotheke Zurich) i.p. as pre-emptive analgesia. Primary tumors were resected, and wounds were closed using Autoclip wound clips (BD Biosciences). For post-operative analgesia 0.1 mg/kg buprenorphine (Temgesic, Schering-Plough) was given i.p. immediately after surgery and in the drinking water at 10 µg/ml for 48 h ad libitum.

For depletion of CD8⁺ T cells, 500 µg rat anti-mouse CD8 (clone YTS 169.4, hybridoma originally obtained from H. Waldmann, Oxford, United Kingdom) was injected intraperitoneally (i.p.) in PBS as described in each experiment. Control mice were injected with 500 µg rat anti-Trinitrophenol (clone 2A3, BioXCell). Antibodies were purified from hybridoma culture supernatant using protein G Sepharose 4 Fast Flow (Sigma). Injection of the respective antibody resulted in depletion of >90% of CD4⁺ or CD8⁺ T cells for at least 14 days, as determined by

flow cytometry. Antibodies were administered i.p. in 200 µl PBS. Full depletion of the targeted population was confirmed by flow cytometry on blood 2 d after injection of the antibody in every experiment.

For blockade of PD-1, mice were injected with 250 µg anti-PD-1 (clone RMP1-14, made in-house by H. Yagita) as indicated. Control mice received 250 µg rat-anti-trinitrophenol (clone 2A3, BioXCell).

For blockade of INFγ and TNFα, mice were injected every 3rd day starting on day 3 until the endpoint with 500 µg anti-INFγ (clone R4-6A2, BioXCell) plus 500 µg anti-TNFα (clone XT3.11, BioXCell) in 150 µl PBS. Control mice received 1000 µg rat-anti-trinitrophenol (clone 2A3, BioXCell).

Lung metastasis from luciferase- or mCherry-expressing tumors was quantified using an IVIS200 imaging system (PerkinElmer) as previously described[23]. Briefly, for luciferase-expressing tumors, mice were injected i.p. with 150 mg/kg D-luciferin (Promega) and photon flux was measured 20 min later in vivo as well as from dissected lungs. Lung metastasis from parental tumors was quantified by India Ink as previously described[23]. Briefly, mice were euthanized and India Ink (Pelikan, 15% in PBS) was injected intratracheally, lungs were harvested, washed in PBS and fixed in Fekete's solution (62% ethanol, 3.3% formaldehyde, 0.25 M acetic acid). Metastatic foci were counted blinded using a dissection microscope.

Disseminated cancer cells were detected in the lungs using a colony-forming assay as previously described[21]. Briefly, lungs were resected from euthanized mice,

cut into small pieces and digested for 45 min at 37 ℃ in DMEM with 1 mg/ml collagenase IV and 2.6 µg/ml DNase I (both Sigma) on a rotating device. Subsequently, samples were washed with PBS by centrifugation at $350 \times g$. For detection of circulating cancer cells, blood was collected by heart puncture in a 25-gauge syringe containing 100 µl heparin (5000 IU/ml, Braun) and cells were washed with PBS by centrifugation at $350 \times g$. Blood and lung samples were suspended in complete DMEM containing 6 µM of 6-thioguanine and cultured in a T175 flask. Medium was exchanged after 5–7 days and presence of colonies of tumor cells was evaluated after 14 days by light microscopy and crystal violet staining.

**Immunofluorescence of mouse samples.** Organs were fixed with 4% paraformaldehyde (Roti-Histofix 4%, Roth) for 10 min at RT and cryoembedded in Optimal Cutting Temperature (O.C.T.) Compound (O.C.T.$^{TM}$ Compound, Tissue-Tek) using dry ice/100% ethanol slurry. Ten-µm thick sections cut using a cryotome (Leica), mounted on SuperFrost glass slides, dried for 1 h at 37 ℃ and preserved at −80 ℃ until immunostained. To prevent unspecific binding of antibodies, slides were incubated with 4% donkey serum in PBS (ANAWA Bio-World) for 10 min at RT. Subsequently, slides were incubated overnight at 4 ℃ with primary antibody diluted in PBS 1% donkey serum. Following primary antibodies were used: Anti-mCherry (goat polyclonal antibody, AB8181-200 SIC-GEN, 1:400), Ki67 (rabbit monoclonal antibody, ab16667 abcam, 1:100). After incubation, slides were washed three times with PBS Tween 20 0.05% and incubated for 1 h at room temperature with secondary antibodies diluted in PBS 1% donkey serum. Following secondary antibodies were used: AF594-conjugated anti-goat IgG, AF488-conjugated anti-rabbit IgG (both Jackson ImmunoResearch, 1:400). Finally, slides were washed, incubated with 0.5 µg/ml 4′,6 diamidine-2-phenylindole (DAPI; Invitrogen, 1:5000) for 5 min, washed again and mounted with ProlongDiamond medium (Invitrogen). The slides were scanned using the automated multispectral microscopy system Vectra 3.0 (PerkinElmer). An unstained slide was used to generate the spectral profile of autofluorescence in studied tissues. The Inform software (PerkinElmer) was used for spectral unmixing of individual fluorophores and autofluorescence.

**Flow cytometry.** Animals were euthanized by isoflurane overdose. The right heart ventricle was perfused with 10 ml PBS to eliminate the blood from the lung vessels. Primary tumors and lungs were collected in DMEM, cut into small pieces and digested for 45 min at 37 ℃ in DMEM containing 1 mg/ml collagenase IV and 2.6 µg/ml DNase I (both Sigma) on a rotating device. Samples were washed with PBS by centrifugation for 5 min at $350 \times g$, the pellet was suspended in PBS and filtered through a 70-µm filter (BD Biosciences) to obtain a single cell suspension. For lymphocyte analysis, cells were further purified by centrifugation over a Percoll gradient (GE Healthcare, 17-0891-01, Sigma Aldrich).

Single cells were stained according to standard protocols. Briefly, cells were surface-stained in 50 µl antibody-mix in PBS. For intracellular cytokine staining, cells were stimulated with 100 ng/ml phorbol 12-myristate 13-acetate (PMA) plus 1 µg/ml ionomycin for 4 h at 37 ℃ in the presence of GolgiPlug/GolgiStop (BD Pharmigen). Cells were stained for surface molecules as described above, washed with PBS, and fixed for 30 min on ice using IC Fixation Buffer from Foxp3/Transcription Factor Staining Buffer Set (eBioscience). Subsequently, cells were stained for intracellular cytokines in permeabilization buffer from the Foxp3/Transcription Factor Staining Buffer Set overnight at 4 ℃. After washing with permeabilization buffer, samples were suspended in FACS buffer (PBS, 20 mM EDTA pH 8.0, 2% FCS) and acquired on a CyAn ADP9 flow cytometer (Beckman Coulter), FACS LSRII Fortessa or FACSymphony (both BD Biosciences). For quantitative analysis, CountBright absolute counting beads were used (ThermoFisher Scientific). In all staining, dead cells were excluded using Live/Dead fixable staining reagents (Invitrogen), and doublets were excluded by FSC-A versus FSC-H and SSC-A versus SSC-H gating. Following directly labeled anti-mouse primary antibodies were used: Anti-CD8a in BUV 805 (clone 53-6.7, rat IgG2a, BD Pharmigen, 1:100), anti-CD11b in BUV 661 (clone M1/70, rat IgG2b, BD Pharmigen, 1:400), anti-CD45.2 in BUV 653 (clone 30-F11, rat IgG2b, BD Pharmigen, 1:400), anti-VISTA in AF488 (clone MH5A, armenian hamster IgG1, BioLegend, 1:100), anti-CD39 in PerCP-eFluor710 (clone 24DMS1, rat IgG2b, ThermoFisher Scientific, 1:100), anti-LAG3 in BV 421 (clone C9B7W, rat IgG1, BioLegend, 1:100), anti-CD44 in BV 570 (clone IM7, rat IgG2b, BioLegend, 1:100), anti-CD73 in BV 605 (clone TY/11.8, rat IgG1, BioLegend, 1:100), anti-CD25 in BV 650 (clone PC61, rat IgG1, BioLegend, 1:100), anti-PD-1 in BV 785 (clone 29F.1A12, rat IgG2a, BioLegend, 1:100), anti-TCRβ in PE-Cy5 (clone H57-597, armenian hamster IgG1, BioLegend, 1:400), anti-KLRG1 in APC-Cy7 (clone 2F1/KLRG1, armenian hamster IgG1, BioLegend, 1:100), anti-TIM3 in AF647, clone B8.2C12, rat IgG1, BioLegend, 1:100), anti-CD103 in Biotin, clone 2E7, armenian hamster IgG1, BioLegend, 1:100), anti-Streptavidin in BUV 395 (BD, 1:200), anti-Ki67 in BV 480 (clone B56, mouse IgG1, BD, 1:100), anti-TNFα in BV 711 (clone MP6-XT22, rat IgG1, BioLegend, 1:400), anti-INFγ BUV 737 (clone XMG1.2, rat IgG1, BD, 1:100), anti-CD4 in BUV 496 (clone GK1.5, rat IgG2b, BD, 1:200), anti-FOXP3 in PE (clone FJK-16s, rat IgG2a, ThermoFisher Scientific, 1:200), anti-EOMES in PE-eFluor610 (clone Dan11mag, rat IgG2a, ThermoFisher Scientific, 1:200), anti-T-bet in PE-Cy7 (clone eBio4B10, rat IgG1, ThermoFisher Scientific, 1:100), anti-CTLA4 in APC-AR700 (clone UC10-4F10-11, armenian hamster IgG1, BD, 1:100) and anti-CD24 in FITC (clone M1/69, rat IgG2b, BioLegend, 1:200).

Flow cytometry data were compensated and exported using FlowJo software (version 10, TreeStar Inc.). The exported FCS files were normalized using Cyt3 MATLAB (version 2017b) and uploaded into Rstudio (R software environment, version 3.4.0). tSNE and FlowSOM algorithm mapping live cells from a pooled sample were performed as described[74]. CellCNN was run using default parameters, dividing data into training and validation steps as described[33].

**Purification of CD39$^+$PD-1$^+$CD8$^+$ T cells.** Primary tumors were processed as described under "Flow Cytometry". After the Percoll gradient, the leukocyte fraction was enriched for CD8$^+$ T cells using anti-mouse CD8a microbeads (Miltenyi Biotec) and autoMACS equipment (Miltenyi Biotec) according to the manufacturer's instructions. Subsequently, cells were stained for CD44, CD39, PD-1, and CD8 as described under "Flow cytometry" and live, single CD44$^+$CD39$^+$PD-1$^+$CD8$^+$ T cells, as well as other CD44$^+$CD8$^+$ T cells, were sorted using an ARIA III Sorter (BD Biosciences).

**Adoptive transfer of CD39$^+$PD-1$^+$CD8$^+$ T cells.** Sorted cells were counted using trypan blue (Trypan Blue solution, Sigma Aldrich) to exclude dead cells. Two hundred-thousand CD44$^+$CD39$^+$PD-1$^+$CD8$^+$ T cells were injected immediately after $10^5$ 4T07-LZ cells. Injections were given intravenously.

**In vitro induction of senescence.** Twenty-thousand 4T07 cells were plated in a 6-well plate. Subsequently, 75 ng/ml mouse IFNγ (R&D Systems) plus 5 ng/ml mouse TNFα (PeproTech) were added and cells were incubated at 37 ℃ in a humidified atmosphere with 5% $CO_2$ for 6 days as described. The proportion of senescent cells was determined by staining for β-galactosidase activity using Senescence β-Galactosidase Staining Kit (9860, Cell Signaling Technology) as described[42].

**In vitro cytotoxicity assay.** CD44$^+$CD39$^+$PD-1$^+$CD8$^+$ T cells were sorted from 4T07 tumors as described above. T cells were cultured at 37 ℃ in a humidified atmosphere with 5% $CO_2$ with mouse T-Activator CD3/CD28 beads (Dynabeads, ThermoFisher) plus recombinant mouse IL-2 (100 U/ml, eBioscience). Cultures were performed in 96-well round-bottom plates with 50,000 T cells and (if applicable) 50,000 activator beads per well. Subsequently, 10,000 live 4T1 or 4T07 cells were incubated with 50,000 live T cells (or medium as control) per well of a 96-well round-bottom plate for 6 h at 37 ℃ in a humidified atmosphere with 5% $CO_2$. Co-cultures were collected and stained with anti-mouse CD24-FITC (clone M1/69, 1:200), anti-mouse CD45.2-APC (clone 104, 1:200), anti-mouse CD8a-BV421 (clone 53-6.7, 1:200) and Zombie-NIR fixable dye (all BioLegend) and samples were acquired on a LSRII Fortessa flow cytometer (BD). T cells were identified as CD45.2$^+$CD8$^+$CD24$^-$, 4T1 and 4T07 target cells as CD45.2$^-$CD8$^-$CD24$^+$. Data were analyzed using FlowJo v10 software (Tree Star). The percentage dead (Zombie-NIR$^+$) target cells was determined after gating on CD45.2$^-$CD8$^-$CD24$^+$ cells.

**RNA-sequencing**

*Library preparation.* The libraries were prepared following the Smart-seq2 protocol[75]. Five-hundred pg of cDNA from each sample were tagmented and amplified using Illumina Nextera XT kit. The resulting libraries were pooled, double-sided size selected (0.5× followed by 0.8× ratio using Beckman Ampure XP beads) and quantified using an Agilent 4200 TapeStation System. The pool of libraries was sequenced in an Illumina NovaSeq6000 sequencer (single-end 100 bp) with a depth of around 20 Mio reads per sample as a service by the Functional Genomics Center Zurich (FGCZ).

*Data evaluation.* The raw data generated by Illumina NovaSeq6000 sequencer were analyzed using SUSHI, a framework for analysis of NGS data developed by the FGCZ[76,77]. Briefly, after quality control, the reads were aligned to a reference genome (Ensembl genome Mus_musculus. GRCII Fortessa dated 2018.02.26) with STAR 2.7.3a. The software package EdgeR was used to detect differentially expressed genes. We applied a threshold of $p < 0.01$, FDR < 1% and a log fold-change of >1.0 for upregulated genes and < −1.0 for downregulated genes. Unsupervised hierarchical clustering was done using the Ward2 method. Heatmaps were generated with the software FunRich V3.1.3.

Gene set enrichment analysis[43,78] was performed on a list of genes ranked from high to low estimated fold-change using the GSEA 4.0.3 Software (Broad Institute) with enrichment statistic classic and 1000 permutations.

**Patient material.** Tumor tissues from 54 patients with breast cancer (Supplementary Table 1) were collected at the University Hospital Zurich, Switzerland. The cohort was established with the intention to eventually compare the immune infiltrate among primary breast cancer tissue and intrapatient matched distant metastasis sites. To this end, we searched breast cancer patients suffering from either invasive-ductal or invasive-lobular breast cancer with hematogenous metastases in the archives of the Department of Pathology and Molecular Pathology, University Hospital Zurich. Our cohort is, therefore, biased for patients with advanced metastatic disease and differs from an average breast cancer cohort. While we cannot formally exclude any bias by this cohort

selection we are not aware of any studies systematically comparing the tumor immune microenvironment in primary breast cancer between metastatic and non-metastatic cases. Donors provided written, informed consent to tissue collection, analysis and data publication according to the Declaration of Helsinki. Law abidance was reviewed and approved by the ethics commission of the Canton Zurich (BASEC-Nr. 2018-02282 and KEK-ZH-2013-0584). Samples were numerically coded to protect donors' rights to confidentiality and privacy.

**Immunofluorescence of human samples**. Formalin-fixed paraffin-embedded samples were cut into 2 μm-thick sections and dried overnight at 55 °C. Antigen retrieval and deparaffinization were performed in 1× Trilogy Solution (Cell Marque, 920P-06) for 15 min at 115 °C. Staining was performed employing tyramide signal amplification (TSA) and the Opal™ 7-color Manual IHC Kit (PerkinElmer). To prevent unspecific binding of antibodies, slides were incubated with 4% donkey serum in PBS (ANAWA BioWorld) in PBS/0.1% Triton X-100 for 15 min at 37 °C. TSA staining protocol was performed as described[79]. Briefly, slides were incubated overnight at 4 °C with primary antibody diluted in PBS/1% donkey serum/0.1% Triton X-100. Subsequently, slides were washed in PBS Triton X-100 0.1% and incubated with corresponding HRP-conjugated secondary antibodies. Following primary antibodies were used: Anti-CD39 (mouse monoclonal antibody, clone A1, BioLegend, 1:500), anti-PD-1 (rabbit monoclonal antibody, clone D4W2J BioConcept, 1:4000), anti-CD8 (mouse monoclonal antibody, clone RPA-T8, Cell Signaling, 1:2000), anti-PAN-Cytokeratin (rabbit polyclonal antibody, cat. no. H-240, Santa Cruz, 1:2000) and anti-EpCAM (rabbit monoclonal antibody, clone EPR20532-225, Abcam, 1:200). Following HRP-conjugated secondary antibodies were used: anti-mouse IgG (H + L) (donkey polyclonal antibody, 715-035-151 Jackson ImmunoResearch 1:500) and anti-rabbit IgG (H + L) (donkey polyclonal antibody, 715-035-152 Jackson ImmunoResearch 1:1000). Fluorescent signal was developed using the Opal™ 7-color Manual IHC Kit. Finally, slides were washed, incubated with 0.5 μg/ml 4′,6-diamidine-2-phenylindole (DAPI; Invitrogen) for 5 min, washed again and mounted with ProlongDiamond medium (Invitrogen). The slides were scanned using the automated multispectral microscopy system Vectra 3.0 (PerkinElmer). An unstained slide was used to generate the spectral profile of autofluorescence in studied tissues. The Inform software (PerkinElmer) was used for (i) unmixing the spectra of individual fluorophores and autofluorescence, (ii) performing automated tissue and cell segmentation into tumor and non-tumor cells, and (iii) exporting single-cell intensity values for all fluorescent signals. Single-cell parameter files were transformed into.fcs files using R script (version 3.6.1) involving FlowCore and Biobase packages and were analyzed using FlowJo software to gate on CD39+PD-1+CD8+ T cell populations in tumor tissue category (Supplementary Fig. 11f). Gates were set based on the intensity values identified in the fluorescent images. Cell density values were measured as number of cells/mm² obtained as cell counts from the gated populations divided by the tumor tissue area measured in the corresponding patient. Samples were analyzed blinded.

**Statistical analysis**. Data were analyzed using GraphPad Prism 8.0 for Windows or Mac, and RStudio (version 1.2.5019). For comparison of 2 experimental groups, 2-tailed Student's $t$ test with Welch's correction was performed. More than 2 groups were compared using an ANOVA test with Bonferroni's correction. Data following a logarithmic distribution (i.e., IVIS signal) were log transformed prior to analysis. The number of mice per experimental group was determined based on our previous experience with similar models. Every point represents one mouse. Unless stated otherwise, data are shown as mean ± SD. $p < 0.05$ is considered significant throughout. $*p < 0.05$, $**p < 0.01$, $***p < 0.001$, $****p < 0.0001$. CellCnn data were analyzed using the Kolmogorov–Smirnov two-sample statistical test. In addition, the population identified by the algorithm was compared in the two groups using the two-tailed Student's $t$ test with Welch's correction. Survival was compared for patients with high and low density of cells to be compared as indicated for the individual graphs by Kaplan–Meier curves. Confidence intervals are indicated as shaded areas surrounding survival curves. Significance was calculated by log-rank test. Patient numbers at risk are displayed for each 15 years of follow-up. The threshold of cell type density was identified using the Receiver Operating Characteristics (ROC) curve analysis.

Density of CD39+PD-1+CD8+ T cells was correlated with the clinicopathological information of the patients using corresponding R packages. Disease-free survival was analyzed by Kaplan–Meier curve and log-rank test (Survminer and Survival R packages), as well as univariate and multivariate Cox regression (pROC, ROCR, and Survival R packages). The threshold of CD39+PD-1+CD8+ or CD8+ T cell density was identified by the receiver operating characteristics (ROC) curve analysis using 3-year disease-free survival for defining long- and short-term survivors. Following values apply: Intra-tumoral CD39+PD-1+CD8+ T cells, 4.20; extra-tumoral CD39+PD-1+CD8+ T cells, 8.43; total CD39+PD-1+CD8+ T cells, 5.71; intra-tumoral CD8+ T cells, 249.20; extra-tumoral CD8+ T cells, 1218.60; total CD8+ T cells, 562.29.

**Reporting summary**. Further information on research design is available in the Nature Research Reporting Summary linked to this article.

## Data availability
RNA-sequencing data that support the findings of this study have been deposited in NCBI with the accession code PRJNA609233. The remaining data are available within the Article, Supplementary Information or available from the authors upon request.

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

## Acknowledgements
We thank Anne Müller, Lubor Borsig, and Christian Münz (all University of Zurich) for valuable input and support. We thank Fred Miller (Karmanos Cancer Institute) for providing cell lines. We thank the personnel from the Zurich Integrative Rodent Physiology (ZIRP, University of Zurich) and the Laboratory Animal Services Center (LASC, University of Zurich) for expert animal care. We thank Ruben Casanova (University of Zurich) for developing the R script converting immunofluorescence image data into file format compatible with flow cytometry data analysis software. This work was financially supported by the Swiss National Science Foundation (SNSF; CRSII5_177208 and 310030_175565 (MvdB), as well as 310030_170320, 310030_188450 and CRSII5_183478 (B.B.)), the Swiss Cancer League (Oncosuisse; KLS-4098-02-2017 (M.v.d.B.) and KFS-4431-02-2018 (B.B.)), the University of Zurich Forschungskredit (P.T.d.L.), the University Research Priority Program (URPP) Translational Cancer Research (M.v.d.B., B.B.), the Science Foundation for Oncology (SFO; (M.v.d.B.)), the Helmut-Horten Foundation (M.v.d.B.), the Hartmann-Müller Foundation Zurich (M.v.d.B.), the Monique-Dornonville-de-la-Cour Foundation Zurich (M.v.d.B.) and the European Union H2020 Project iPC #826121 (B.B.).

## Author contributions
P.T.d.L., H.C., and M.v.d.B. conceived experiments; P.T.d.L., H.C., M.V., N.N., P.C., E.M.C., V.C., K.S., F.M.A., J.U., I.G., M.P.L., and S.H. performed and analyzed experiments; P.T.d.L., H.C., B.B., and M.v.d.B. wrote the manuscript; B.B., S.T., B.S., H.Y., and H.M. provided essential reagents; B.B. helped with data analysis; P.T.d.L. and M.v.d.B. secured funding. P.T.d.L. and H.C. contributed equally. N.N. and M.V. contributed equally.

## Competing interests
The authors declare no competing interests.
