## [Peer Review File · Nature Communications]

Reviewers' Comments:

Reviewer #2:

Remarks to the Author:

In the present study employing 4T1 and 4T07 breast cancer model de Lara and colleagues report CD39+PD1+CD8+ T-cells mediate metastatic dormancy in breast cancer. Authors demonstrate CD39+PD1+CD8+ T-cells aid tumor cell dissemination and dormancy in 4T07 model. Overall, this study presents interesting observations regarding CD39+PD1+CD8+ T-cell function but has several limitations listed below.

Major Points:

Infiltration of CD39+PD1+CD8+ T-cells is high in 4T07 tumors and lungs of mice bearing 4T07 tumors compared to 4T1. CD39+PD1+CD8+ T-cells secrete TNF- α and IFN- γ that attenuates proliferation of both 4T1 and 4T07. The inability of CD39+PD1+CD8+ T-cells isolated from 4T07 tumor bearing mice to limit 4T1 metastasis is interesting. 4T1 could be inherently aggressive as authors suggest or there are significant differences in infiltrating CD39+PD1+CD8+ T-cell population in lungs compared to 4T07/4T1 primary tumors. No difference between CD39+PD1+CD8+ T-cells from primary tumor of 4T1 and 4T07 could also imply no direct role in promoting metastatic dormancy. Likewise, the inability of 4T07 to metastasize to lungs at higher rates comparable to 4T1 in the absence of T-cells (Foxn1^{nu/nu}) indicate cell-intrinsic mechanisms of dormancy are dominant.

How CD39+PD1+CD8+ T-cells in the lungs enforce dormancy in 4T07 cells and how do 4T07 cells induce CD39+PD1+CD8+ T-cell infiltration and response is interesting and not addressed. No direct evidence showing presence of CD39+PD1+CD8+ T-cells in proximity of dormant 4T07 cells in lungs is provided. Profiles of lung and primary tumor infiltrating CD8+ T-cell populations are likely to provide much needed key insights.

CD8+ T-cell memory generated by 4T07 primary tumor protects against 4T1 metastasis but adoptive transfer of CD39+PD1+CD8+ T-cells does not limit 4T1 lung metastasis even though both 4T07 and 4T1 cells are susceptible to CD39+PD1+CD8+ T-cells in vitro is not clear and needs to be addressed.

The effect of anti-TNF- α and IFN- γ antibodies on promoting 4T07 metastatic outbreaks is not clear. Clear images showing metastatic lesions/micro-metastases, increased BLI signal and detailed quantification of single cells and metastatic outbreaks events is needed. Does TNF- α and IFN- γ induce senescence in 4T1 cells or is it exclusive to 4T07? Could provide mechanistic insights.

Minor points:

Quantification of number of disseminated 4T07 cell in mouse lungs with primary tumor and upon resection is needed for 1F, similar to 1C.

As controls quantification of total CD8+ cells and CD39+PD1+CD8+ T-cells in primary as well as lungs on day 10/11 (4T1 and 4T07) is needed to justify priming effects of CD39+PD1+CD8+ T-cells before I.V. challenge.

Images lacking clarity and needs replacement - Figure 1D all channels 4T07-mch-DCC in lungs; Extended Figure 1F all channels 3rd from the top; Extended figure 3D - out of focus. Also, please check labeling for Figure 2K, Figure 5B, and Extended Figure 7D.

Reviewer #3:

Remarks to the Author:

The authors have taken into consideration my previous comments, performed further work, generated new experimental data and revised their manuscript accordingly. The amended and updated manuscript clearly presents the message of the study. I think, in the present form, the manuscript merits consideration for publication in Nature Communications.

Point-by-point reply to Reviewer #2

In the present study employing 4T1 and 4T07 breast cancer model de Lara and colleagues report CD39+PD1+CD8+ T-cells mediate metastatic dormancy in breast cancer. Authors demonstrate CD39+PD1+CD8+ T-cells aid tumor cell dissemination and dormancy in 4T07 model. Overall, this study presents interesting observations regarding CD39+PD1+CD8+ T-cell function but has several limitations listed below.

We thank the reviewer for these helpful comments and reply to all points below. Our reply is in green and the **changes made in the manuscript are in bold**.

Because of novel data, the Supplementary Figures have been relabeled. The numbers in our reply relate to the final labeling according to the final manuscript text.

Major Points:

Infiltration of CD39+PD1+CD8+ T-cells is high in 4T07 tumors and lungs of mice bearing 4T07 tumors compared to 4T1. CD39+PD1+CD8+ T-cells secrete TNF-alpha and IFN-gamma that attenuates proliferation of both 4T1 and 4T07. The inability of CD39+PD1+CD8+ T-cells isolated from 4T07 tumor bearing mice to limit 4T1 metastasis is interesting. 4T1 could be inherently aggressive as authors suggest or there are significant differences in infiltrating CD39+PD1+CD8+ T-cell population in lungs compared to 4T07/4T1 primary tumors. No difference between CD39+PD1+CD8+ T-cells from primary tumor of 4T1 and 4T07 could also imply no direct role in promoting metastatic dormancy. Likewise, the inability of 4T07 to metastasize to lungs at higher rates comparable to 4T1 in the absence of T-cells (Foxn1^{nu/nu}) indicate cell-intrinsic mechanisms of dormancy are dominant.

We indeed show by RNA-Seq that the CD39+PD-1+CD8+ T-cells from 4T1 and 4T07 tumors are very similar. The reason why these cells do not induce dormancy via IFN-gamma and TNF-alpha in the case of 4T1 orthotopic tumors is simply because there are hardly such CD39+PD-1+CD8+ T-cells in 4T1 tumors.

We have shown multiple lines of evidence for an essential role of CD39+PD-1+CD8+ T-cells in induction of metastatic dormancy of 4T07:

1. CD8-depletion of mice with orthotopic 4T07 breast cancer results in progressive lung metastasis (Figures 2a-c).
2. Orthotopic 4T07 induces progressive lung metastasis in T-cell-deficient nu/nu mice (Figure 2c, 2e).
3. Intravenous injection of 4T07 results in progressive lung metastasis in naïve mice (Figures 2a, 2b), but in metastatic dormancy in 4T07-immune mice (Figure 2i-k), unless the latter were CD8-depleted (Figures 2l, 2m).
4. Adoptive transfer of purified CD39+PD-1+CD8+ T-cells induces dormancy of i.v. injected 4T07 cells after their dissemination to the lungs (Figures 4c-f). In these experiments we didn't visualize the dormant cells, thus leaving some room for the possibility that the i.v. injected cells actually never reached the lungs when CD39+PD-1+CD8+ T-cells were transferred. For this reason, we already performed the same experiment and visualized disseminated 4T07 cells in the lungs. We see metastatic outgrowth in all lungs of mice that did not receive CD39+PD-1+CD8+ T-cells; in contrast, in mice that received CD39+PD-1+CD8+ T-cells, we found single, non-proliferating 4T07 (i.e. dormant) cells in the lungs. **These new data are incorporated in the final revision as Supplementary Figure 8a-c and discussed on page 11 of the manuscript.**

Taken together, we have shown that CD39+PD-1+CD8+ T-cells primed by a 4T07 breast tumor are necessary and sufficient to mediate metastatic dormancy in the lungs.

The fact that 4T1 cells are intrinsically more metastatic than 4T07 cells in immunodeficient mice reflects (a combination of) traits that are different between 4T1 and 4T07 cells, some of which are unrelated to T-cells. Although we discovered this population by comparing the tumor microenvironment of 4T1 (progressively metastatic) and 4T07 (dormant) orthotopic breast cancer, the comparison of these two cell lines in any sense is definitively not the purpose of our work. In fact, we think that the many differences between 4T1 and 4T07 cells preclude appropriate and conclusive comparison *in vivo*, as the reviewer suggested.

We have included a paragraph better explaining this on page 6 of the manuscript.

How CD39+PD1+CD8+ T-cells in the lungs enforce dormancy in 4T07 cells and how do 4T07 cells induce CD39+PD1+CD8+ T-cell infiltration and response is interesting and not addressed. No direct evidence showing presence of CD39+PD1+CD8+ T-cells in proximity of dormant 4T07 cells in lungs is provided. Profiles of lung and primary tumor infiltrating CD8+ T-cell populations are likely to provide much needed key insights.

We showed that CD39+PD1+CD8+ T-cells enforce metastatic dormancy through TNF- α and IFN- γ (Figures 4g-h and Supplementary Figure 9). We did not address why 4T07 cells induce more CD39+PD1+CD8+ T-cells than 4T1 because our aim was not to explain the differences between these two cell lines (which is likely complex and multifactorial, as described in different publications such as Lu et al. J Biol Chem 2010, Tien Vu et al. J Extracell Vesicles 2019, Bemmo et al. PLoS One 2010) but rather to better understand the process of metastatic dormancy.

Figure 3 shows extensive profile of CD39+PD1+CD8+ T-cells in the primary tumors and lungs. Moreover, we also show that these cells infiltrate primary human breast cancers in Figure 6. We have not shown co-localization of CD39+PD1+CD8+ T cells with dormant DTCs, because we think that this would not provide any additional evidence to that already shown in Figures 4 and 2.

CD8+ T-cell memory generated by 4T07 primary tumor protects against 4T1 metastasis but adoptive transfer of CD39+PD1+CD8+ T-cells does not limit 4T1 lung metastasis even though both 4T07 and 4T1 cells are susceptible to CD39+PD1+CD8+ T-cells *in vitro* is not clear and needs to be addressed.

We show in Supplementary Figure 4c-d that 4T07-induced CD8+ T-cell memory indeed partially protects against 4T1 metastasis. However, we would like to specify this: Although the lung metastatic load resulting from *i.v.* injected 4T1 cells is reduced in 4T07-bearing mice, the lesions are still progressive (*i.e.* multicellular and Ki67+) and not dormant (Supplementary Figures 4a-d). **We have included a paragraph better explaining this on page 8 of the manuscript.**

Furthermore, we think that the number of adoptively transferred CD39+PD-1+CD8+ T-cells is simply insufficient to control 4T1 cells that have an intrinsically higher metastatic potential.

The effect of anti-TNF- α and IFN- γ antibodies on promoting 4T07 metastatic outbreaks is not clear. Clear images showing metastatic lesions/micro-metastases, increased BLI signal and detailed quantification of single cells and metastatic outbreaks events is needed. Does TNF- α and IFN- γ induce senescence in 4T1 cells or is it exclusive to 4T07? Could provide mechanistic insights.

Orthotopic 4T07 tumors never induce progressively growing metastatic lesion in the lungs; instead, we observe single, non-proliferating disseminated cells in the lungs of all mice analyzed. Thus, the mere presence of multicellular metastatic 4T07 nodules in the lungs after blocking TNF α and IFN γ unequivocally shows that induction of dormancy essentially depends on these two cytokines (Figures

4g, 4h). We did not quantify metastatic lesions in the lungs of mice that received IFN γ - and TNF α -blocking antibodies, since already their presence provides a conclusive answer. **We have included a paragraph better explaining this on page 12 of the manuscript.**

IFN γ and TNF α did not induce senescence but rather cell death of 4T1 cells in vitro. This finding may be interesting but is beyond the scope of this project, because (i) the T-cell-derived cytokines are unlikely to be present at a sufficient concentration in the lungs of 4T1-bearing mice, simply because the producing cells (CD39+PD-1+CD8+ T-cells) are largely absent, and (ii) our main message is not the comparison of 4T1 and 4T07 as explained above.

Minor points:

Quantification of number of disseminated 4T07 cell in mouse lungs with primary tumor and upon resection is needed for 1F, similar to 1C.

We have quantified the DTC in Figure 1f and **added these data as a new panel to Figure 1 as panel f**; because of this, the original panel 1f is now relabeled as 1g.

As controls quantification of total CD8+ cells and CD39+PD1+CD8+ T-cells in primary as well as lungs on day 10/11 (4T1 and 4T07) is needed to justify priming effects of CD39+PD1+CD8+ T-cells before I.V. challenge.

We have incorporated the data showing that the number of CD39+PD1+CD8+ T-cells in the primary tumor is already increased in 4T07 tumors on day 10 in the final revision as new Supplementary Figure 7a-d and discussed the data on page 10 of the manuscript.

Images lacking clarity and needs replacement - Figure 1D all channels 4T07-mch-DCC in lungs; Extended Figure 1F all channels 3rd from the top; Extended figure 3D - out of focus.

We have checked the images carefully and did not understand what the reviewer meant by lacking clarity for Figure 1d and Extended Figure 1f.

We do agree that Supplementary Figure 3d is out focus and we **have replaced it by a better image.**

Also, please check labeling for Figure 2K, Figure 5B, and Extended Figure 7D.

We have carefully checked the labeling of the panels mentioned above. As far as we could see, 2k and 5b are correct. We have **added the missing “d” in the figure** and thank the reviewer for spotting this omission.